# Effects of biobanding on training loads and technical performance of young football players

Jaimar Fellipe Silva de Macedo, Bruno Laerte Lopes Ribeiro◉*, Ayrton Bruno de Morais Ferreira, Ricardo Santos Oliveira, Arnaldo Luis Mortatti

INTEGRA – Integrative Physiology, Health, and Performance Research Group, Federal University of Rio Grande do Norte, Natal, Brazil

* contatobrunoribeiro@outlook.com.br

**Data Availability Statement:** All relevant data are within the paper and its Supporting Information files.

## Abstract

### Background

In adolescent sports, grouping by chronological age can advantage athletes born early in the year due to maturation differences. Early maturing athletes often achieve greater performance gains, are perceived as more talented, and receive more specialized training and workloads. This study aimed to assess the effects of biobanding on internal and external training loads, as well as technical performance during small-sided games (SSGs).

### Methods

Twenty male footballers (11.8 ± 1.15 years) participated in this study. Athletes engaged in small-sided games (SSGs) under two conditions: (1) CA–teams formed based on chronological age; and (2) BIO–teams formed based on age relative to peak height velocity (pre-PHV = -2.5 to -1.5, and PHV = -1.5 to -0.5). External load (ETL) was quantified using the PlayerLoad method, while internal load (ITL) was measured using both training impulse (TRIMP) and session-RPE. Player involvement was determined by summing all technical actions performed during the SSGs, with involvement in the game assessed through video analysis of the sessions.

### Results

BIO games significantly increased ETL for pre-PHV (EM = 415.5; 95%CI = 381.5–449.5 a.u.) compared to the CA games (EM = 388.8; 95% CI = 354.8–422.8 a.u.). PHV players had lower ETL (EM = 354.4; 95% CI = 320.4–388.4 a.u.) in BIO compared to CA games (EM = 366.0; 95% CI = 332.0–400.0 a.u.). No significant ITL differences were observed. BIO lowered steals among pre-PHV players vs. CA.

**Funding:** The author(s) received no specific funding for this work.

**Competing interests:** The authors declare no conflicts of interest.

## Conclusion

Biobanded games significantly increased external load (ETL) for pre-PHV players; however, this increase was not substantial enough to affect internal load (as measured by session-RPE and TRIMP) or player involvement.

## Introduction

In adolescent sports the chronological age of the athletes is used to group teams in competition, such as under 13, under 14, under 15, and so on. However, athletes of the same chronological age could exhibit considerably variation in their maturational status [1]. Consequently, employing an under-a-given-age-system tends to advantage athletes born in the early months of the calendar year, a phenomenon known as the relative age effect [2–4]. In fact, early maturing athletes may achieve greater performance gains and are more likely to be perceived by coaches as more talented. As a result, they may gain access to specialised training, training resources, and investment in their sporting careers [5]. An alternative approach to mitigate individual differences arising from biological maturation within the under-a-given-age-system is the implementation of biobanding.

Biobanding has been proposed to reduce discrepancies resulting from physical and cognitive variation within age groups [6–8]. This method involves grouping young athletes into bands or groups based on the predicted adult height or the age of peak height velocity (aPHV), both which serve as indicators of biological maturation, this division has been considered an interesting way to promote physical training in groups of athletes with different ages and maturational stages, considering that athletes with greater biological maturation tend to obtain physical advantages over their peers who have not yet reached PHV [9–12]. Although advocated for training and competition, few studies have tested the impact of biobanding on training loads, with limited and discrepant evidence available. For example, Towlson et al. [13] found a no effect of biobanding on internal and external training load, although improvement of a considerable number of psychological variables for pre-PHV players was observed in the biobanded condition. On the contrary, Barrett et al. [12] found that pre-PHV players report higher perception of exertion compared to post-PHV players during small-sided games (SSG) using the bioband approach. Altogether, evidence is needed to suggest the use of biobanding for both competition and training [1].

In the context of football training, using methods that facilitate both morphological and functional adaptations and the acquisition of sport-specific skills may prove superior, especially in early years of athletic development [14–16]. In this way SSG have been widely used due to its ability to create specific learning contexts that enhance technical and physical skills in the young athletes [17–19].

During SSG practice, diverse strategies can be used to alter training load, which can be assessed using training impulse metrics, external loads, or the perceived exertion of the session (session-RPE) [20, 21]. However, a characteristic of SSG is to combine athletes with a range of chronological age and biological maturation levels (e.g., 4x4, 5x5 or 6x6 format), which can cause the biologically younger players to experience higher training loads when training with their biologically older peers, furthermore, when teams are divided by biobanding and chronological age, it is clear that differences may occur in technical and physical aspects during sports practice [22, 23]. Thus, using external training load analysis instruments, such as the PlayerLoad method—which quantifies total movement across three axes

(horizontal, vertical, and lateral), offering objective data on mechanical stress and the load imposed during activity. In contrast, the internal load, evaluated through the Session Rating of Perceived Exertion (session-RPE) method, reflects the individual's physiological and psychosocial response to training, considering variables such as fatigue and perceived effort. The combination of these approaches allows for a more holistic assessment and is crucial for properly monitoring adaptations to training and preventing injuries or overload, ensuring a balance between stimulus and recovery in sports planning. By combining external load values with internal load measurements, we can evaluate a player's ability to handle the training session [24]. On the other hand, analyzing technical parameters of the game can provide valuable insights into a player's involvement, depending on their maturational status. Biobanding, in particular, can create more opportunities for late-maturing players to utilize their technical competence effectively [23].

Given that biobanding has been advocated not only for competition but also for training purposes, biobanding can be a strategy for coaches seeking better control over the training loads experienced by their adolescent athletes. Thus, the aim of this study was to compare external and internal loads, as well as technical performance during SSG in 10- to 13-year-old athletes when playing in bioband or chronological age conditions. Given the potential impact of biobanding on training loads [12] and technical actions during SSG practice [23], the main hypothesis of the study is that mature late participants will experience higher external and internal training loads and exhibit poorer technical performance when playing under the CA condition.

## Methods

### Participants

Twenty young footballers (11.8 ± 1.15 years), volunteered for this study. They were recruited based on convenience to meet the inclusion criteria of the study: (A) participation in the under-13 category, (B) training with the group for at least six months, and (C) having attended at least 80% of the training sessions during this period. Three participants were between 9.5 to 10.5-year-old, three were between 10.6 and 11.5-years-old, eight were between 11.6 to 12.5-year-old, and six were between 12.5 to 13.5-years-old. Considering the category and the age groups included the positions on the field have not yet been clearly defined. Every participant originated from the same youth football academy. Although there are some young athletes of younger ages (under 11 years old), they have already trained in this category (U13) due to their physical and technical performance. All players performed the same training routine described in Table 1. Players were guided by the same coach. The players typically participated in two football training sessions per week (strength and conditioning and technical-tactical sessions) at the same time of day, and each training sessions lasted between 60 and 75 minutes. Any player unable to complete the study due to musculoskeletal injuries or health issues was

Table 1. Participants' usual training routine.

| Training Sessions | Performed activities |
|---|---|
| Warm-up (15 min) | Muscle stretching, development of basic motor skills (running, jumping, lateral movements) and basic fundamentals activities. |
| Technical training (30 minutes) | Activities to develop specific football skills, such as: Passes, kicks, headings, tackles and game transitions. |
| Game-specific activity | Small-sided games or tactical activities (Games with rule changes, or the development of specific game situations according to the training session proposal) |

excluded. Participants and their guardians provided written consent and assent, and all procedures were approved by the local Research Ethics Committee.

## Design

This experimental study involved participants performing SSGs under two different conditions: 1) CA—SSGs with players divided according to their chronological age (i.e., Randomized teams without the use of biobanding). The CA group was randomized while ensuring that all age ranges were similarly represented in each of the 4 teams formed. Each team included players aged 10, 11, 12 and 13 years old, ensuring balanced age distribution across all teams; and 2) BIO—SSGs with players divided according to their biological age (i.e., biobanded games). The study was completed over three weeks, with the first week used to obtain participants' characteristics, and the second and third weeks used to perform the SSGs under the experimental conditions, it is important to highlight that each SSGs condition was performed twice for each group evaluated (CA and BIO). For the BIO condition, years from PHV was used to divide participants into two different biobanded groups: 1) group pre-PHV, consisting of participants with years from PHV between -2.5 to -1.5 years; and 2) group PHV, consisting of participants with years from PHV between -1.5 to -0.5 years. This method to divide participants according to their PHV was chosen based on previous investigations [12].

A total of six SSGs were completed, with two of one in the CA condition performed in the second week, and four in the BIO condition (2 times with pre-PHV group and 2 times with PHV group) performed in the third week. Each participant completed four SSGs, two in the CA condition and two in their respective biobanded groups (i.e., pre-PHV vs pre-PHV and PHV vs PHV). In the second- and third weeks, the games were completed with a 48-h interval. During the SSGs, players wore accelerometers to obtain external training load in each game played, and at the end of the SSGs, they reported their rate of perceived exertion of the session (session-RPE). All games were video recorded for subsequent technical performance analysis. The mean of the results for all variables collected during the 2 games conducted under each condition (CA and BIO) was retained for analysis.

## Participants characteristics

Body mass (kg), stature (cm), and sitting height (cm) were measured with an electronic scale and stadiometer (Sanny®, Brazil). Somatic maturation was estimated according to [25] as years from peak height velocity (yPHV) using the following equation:

$$
\begin{aligned}
\text{Maturity offset} ={} & -9.236 + 0.0002708\,(\text{Leg Length x Sitting Height}) \\
& -0.001663\,(\text{Chronological age x Leg Length}) \\
& +0.007216\,(\text{Chronological age x Sitting Height}) \\
& +0.02292\,(\text{Weight/Height}).
\end{aligned}
$$

The countermovement vertical jump (CMJ) was used to assess power of the lower limbs. To maximise jump height, athletes were instructed to begin in an upright position with their hands on their hips and to perform the eccentric and concentric phases of the movement as fast as possible using the stretch-shortening cycle. A contact mat (Jump System Pro; Cefise®, Nova Odessa, SP, Brazil) was used to measure jump height, and the highest value of three attempts was used as the CMJ [26]. The intraclass correlation coefficient between CVJH performance measurements was 0.996.

## Small-sided games

In each experimental condition (BIO and CA), players competed in standardised 5-a-side SSGs with goalkeepers. The SSGs were played in four bouts of five minutes interspersed by three minutes of active recovery. The SSGs were performed on an artificial grass pitch measuring 40m x 25m, with a total of 100m$^2$ per player [27]. No rules were in place to limit the number of passes or goals, allowing players to have more ball contact [28], and facilitating development of technical skills [29]. The ball was promptly placed in the centre after a goal was scored, and quickly replaced when out of bounds. Players were verbally encouraged by staff members and researchers during the SSGs. SSGs were preceded by a 5-min warm up consisting of jogging and stretching.

## External training load

External training load was obtained as PlayerLoad using triaxial accelerometers, as previously described [29, 30]. For this, a waistband with an accelerometer (wGT3X-BT, Actigraph® LLC Pensacola, FL, USA) was attached to the right mid-axillary line of the iliac crest of each athlete during all SSGs. This positioning was chosen because it is close to the players' centre of mass, providing better acceleration data [31]. Each athlete used the same device in the SSGs. Accelerometers were set to record acceleration at 100Hz, and the raw data was downloaded for later analysis (Actlife 6.13.3; Actigraph® LLC Pensacola, FL, USA).

The raw accelerometer data was used to calculate PlayerLoad using a homemade routine in Matlab® R2015a (8.5.0.197613). For this, the vector magnitude was obtained from the values of each of the triaxial axes (x, y, and z) according to the Cartesian formula:

$$\sqrt{\left[(x\,n - x\,n - 1)^2 + (y\,n - y\,n - 1)^2 + (z\,n - z\,n - 1)^2 / 100\right]}$$

Where: x = lateral-lateral axis; y = horizontal axis; z = vertical axis (BOYD et al., 2011).

The PlayerLoad is a method for measuring external training load [29]. It is based on the magnitude of changes in acceleration [32]. This method is positively correlate (r = 0.70) with other measures of external load, such as the total distance covered in a match [33]. The validity and reliability of PlayerLoad in team sports using triaxial accelerometers have been demonstrated previously [34]. Furthermore, PlayerLoad is an indicator that correlates with the session RPE method [(r = .51, p < .001) [35, 36].

## Internal training load

**Rating perceived effort of session (Session-RPE).**   Internal training load was obtained as the session-RPE and training impulse (TRIMP). For this, participants rated their perceived exertion using the 0–10 Borg's scale, modified by [37], 30 minutes following the end of the SSGs. Participants were instructed to considered 10 as the highest physical effort they had experienced, and 0 as a condition of absolute rest. Players were allowed to provide decimals as answers (e.g., 7.5). All participants were familiarized with the CR-10 scale during previous training. In addition, the use of RPE in children and adolescents has been previously described.

**Training impulse (Trimp).**   The method proposed by [38] was used to obtain TRIMP. For this, heart rate (HR) was collected during the SSGs using a Polar® H10 heart rate sensor (Kempele, Finland), and the beat-to-beat HR was sent to the accelerometer for recording. Analyses were performed using Kubios HRV software version 3.3.1. TRIMP was calculated as the sum of the time spent in five heart rate zones multiplied by its respective zone value. Heart rate zones were defined as follows: zone 1 = 50% to 60% of HRpeak, zone 2 = 60% to 70% of

HRpeak, zone 3 = 70% to 80% of HRpeak, zone 4 = 80% to 90% of HRpeak, and zone 5 = 90% to 100% of HRpeak. Peak heart rate (HRpeak) was calculated using the highest value found between maximum heart rate (age-adjusted formula) (Manini, et al., 2001) and the peak values achieved during all sessions [39].

## Technical performance

Athletes' technical performance was obtained using video recordings of the SSGs. Actions indicative of technical performance was quantified using notational analysis in the form of frequency of actions [40, 41]. Notation was done for any type of action with a clear attempt to intervene on the trajectory of the ball was observed, both in terms of maintaining possession or finishing at the opposing goal, as well as actions to recover the ball when defending. Technical actions were classified as previously suggested [42, 43]. An action was counted every time a player performed the following technical actions: pass, shot, tackle, dribble, and steal. All actions were counted as successful or non-successful (e.g., a steal that did not lead to ball possession) and total number of actions (successful + non-successful) was retained for analysis. Player involvement was obtained by adding up all technical actions performed during the SSGs. The definitions of technical actions are **Pass**—transferring ball using body except head, arms, hands, for possession or goal opportunity; **Shot**—aiming at goal with body except hands, arms, head, needing defence or stopping by opponents; **Tackle**—regaining possession via physical contact; **Dribble**—keeping ball via feint or pass; **Steal**—gaining possession without heavy contact. Two researchers analyzed video actions, Kappa >0.95; disagreements consulted third researcher.

## Statistical analysis

Data are presented as mean and standard deviation, unless otherwise stated. For the descriptive data, normality of distribution was tested using the Shapiro Wilk's test and homogeneity of variances tested using the Levine's test. Independent t-tests were used to compare participants characteristics according to maturation levels and when necessary, the non-parametric Mann Whitney test was applied.

The inferential part of the investigation was done into two separated steps. Firstly, we used generalized and linear mixed models (LMMs) to investigate the reliability of the technical and player load variables measured in the first and second SSGs played within each condition. For this, fixed effects were obtained for match (SSG #1 and SSG#2), condition (CA and BIO), and their interaction. From these models we investigated whether changes in SSG performance were evident. We also obtained from these models the intraclass correlation coefficient (ICC). To account for the repeated-measures design, a random-effect was applied for participants. We also added random slopes to the models, which did not improve model predictions according to AIC and BIC criteria.

Following the reliability analysis, we investigated the effects of maturation (pre-PHV and PHV), condition (CA and BIO), and their interaction on training load variables. To account for the repeated-measures design, a random-effect was applied for participants. We also added random slopes to the models which did not improve model predictions according to AIC and BIC criteria. To address a possible effect of the SSG performance changing over time due to the lack of reliability, we also inserted in the models a match variable (SSG#1 and SSG#2) as a confounder in the models. LLMs assumptions were explored visually by inspecting the normal distribution of the residuals, which was also tested using the Shapiro Wilk's test [44].

To investigate the impact of maturation and condition on involvement we used a generalized LMM (GLMM) with a Poisson distribution with a log link function. This was done

**Table 2. Descriptive statistics of participants in each assessed condition.**

|  | All (n = 20) | pre-PHV (n = 10) | PHV (n = 10) | p |
|---|---|---|---|---|
| Age (years) | 11.8 ± 1.15 | 11.0 ± 1.1 | 12.6 ± 0.5 | <0.001* |
| yPHV (years) | -1.47 ± 1.18 | -2.4 ± 0.6 | -0.5 ± 0.7 | <0.001* |
| Height (cm) | 153.64 ± 10.57 | 145.7 ± 6.0 | 161.6 ± 7.6 | 0.004* |
| Body mass (kg) | 51.36 ± 18.14 | 38.7 ± 8.0 | 64.1 ± 16.5 | <0.001* |
| CMJ (cm) | 25.03 ± 5.43 | 26.1 ± 4.8 | 24.0 ± 6.1 | 0.481 |

Values are mean and standard deviation. yPHV: years form peak height velocity;

* Difference from pre-PHV to PHV condition.

because involvement failed to meet the assumptions of LLM and was obtained as a sum of count variables. All other technical variables were not analysed due to their zero-inflated distribution and poor ICC values (<0.20). GLLM was constructed with a random effect for participant and for the slope between conditions.

For all variables, each participant had four observations in the models (i.e., two SSG played in the CA condition and two SSG played in their respective BIO condition) leading to a total of 80 observations. All analyses were performed using R 4.0.3 in RStudio Version 2023.03.0 +386.

## Results

Participants characteristics are presented in Table 2. The final sample was formed by 20 athletes because one participant was excluded due to health issues. After dividing athletes into groups of maturation, 10 athletes were classified as pre-PVH and 10 as PHV. No differences were observed in the CMJ performance between groups of maturation (p<0.05).

### Reliability

When investigating whether there were changes in SSGs within condition (e.g., as an indicative of reliability), it was found that session-RPE (ICC = 0.40) and PlayerLoad (ICC = 0.67) presented low reliability, although no changes were observed between SSGs within conditions. However, TRIMP in addition to presenting low reliability (ICC = 0.18), also significantly changed from SSG #1 to SSG #2. For BIO, TRIMP was 107.88 (95%CI = 101.96–113.80) in SSG#1 and decreased to 92.16 (95%CI = 86.24, 98.08) in SSG #2. For CA, TRIMP was 92.26 (95%CI = 86.33–98.18) in SSG #1 and increased to 98.59 (95%CI = 92.67–104.51) in SSG #2. For the player involvement analysed in the study, the observed ICC for involvement was 0.72 and it did not change from one SSG to the other.

### Effects of biobanding on training load

The effects of biobanding on internal and external training loads for pre-PHV and PHV players are present in Tables 2 and 3. While no effects of biobanding were observed on session-RPE and TRIMP (all estimates with $P > 0.05$), a maturation effect was observed with the PHV players reporting a 1.2 a.u. lower session-RPE compared to the pre-PHV players (95% CI = -2.27–-0.03 a.u., $p$ = 0.043). For the external training load, significant effects for maturation, condition and condition*maturation were observed (Table 4). While for the pre-PHV players, the CA condition led to -26.7 a.u. lower PlayerLoad (95% CI = -48.24–-5.17 a.u., $p$ = 0.017) compared to the BIO condition, for the pre-PHV players the CA condition led to 11.5% higher PlayerLoad (95% CI = -10.38–33.40 a.u.; $p$ = 0.298) (Fig 1).

**Table 3. Internal and external training load obtained from the SSG played in the biobanded and chronological conditions.**

| | pre-PHV (n = 10) | | PHV (n = 10) | |
| --- | --- | --- | --- | --- |
| | BIO-SSGs (n = 2) | CA-SSGs (n = 2) | BIO-SSGs (n = 2) | CA-SSGs (n = 2) |
| Session-RPE (au) | 6.3 (5.5–7.1) | 5.9 (5.1–6.6) | 5.1 (4.4–6.0) | 5.0 (4.2–5.8) |
| TRIMP (au) | 99.4 (92.8–106.1) | 91.1 (84.4–97.8) | 100.6 (94.0–107.3) | 99.8 (93.1–106.4) |
| PlayerLoad (au) | 415.5 (381.5–449.5) | 388.8 (354.8–422.8) | 354.4 (320.4–388.4) | 366.0 (332.0–400.0) |

Data are mixed model linear regression estimated mean and 95% confidence interval. SSG: small-sided game. RPE: rate of perceived exertion.

**Table 4. Effects of condition and maturation on internal and external training loads (arbitrary unit) and involvement during the small-sided games.**

| | Session-RPE | TRIMP | Total PlayerLoad | Involvement** |
| --- | --- | --- | --- | --- |
| Intercept | 6.30 (5.51–7.09) | 99.42 (92.87–105.97) | 415.49 (382.04–448.95) | 25.46 (20.55–31.53) |
| Condition | -0.45 (-1.24–0.34) | -8.34 (-16.83–0.16) | **-26.71 (-48.24–-5.17)*** | 0.89 (0.66–1.20) |
| Maturation | **-1.15 (-2.27–-0.03)*** | 1.20 (-8.07–10.47) | **-61.06 (-108.37–-13.74)*** | 1.00 (0.74–1.35) |
| Interaction | 0.30 (-0.81–1.41) | 7.48 (-4.53–19.48) | **38.21 (7.76–68.66)*** | 1.18 (0.77–1.81) |
| **Random Effects** | | | | |
| Residuals SD σ2 | 1.27 | 13.70 | 34.74 | 0.04 |
| Intercept SD τ00 | 0.901 | 4.239 | 48.07 | 0.10 |
| Slope SD τ11 | - - | - - | - - | 0.20 |
| ICC | 0.33 | 0.09 | 0.66 | 0.72 |
| N | 20 | 20 | 20 | 20 |

Data are estimates and 95% confidence interval. Intercept estimates are mean values for pre-PHV players in the biobanded condition. Condition estimates are the changes in mean from the intercept for the chronological condition. Maturation estimates are the changes in mean from the intercept for the PHV players. Interaction (condition*maturation) estimates are the additional changes in mean for PHV players in the chronological condition.

*Bold values indicate significant effects at $P<0.05$. For exact $P$-values please see text. ICC: intraclass correlation obtained from LMMs.

**Estimates are incidence rate ratio from the Poisson GLMM (except the intercept) with random effects for the intercept and slopes.

### Effects of biobanding on technical involvement

The statistical values regarding the technical involvement of the young athletes in both conditions are presented in Table 3. No effects of biobanding or maturation on involvement were observed, as all estimates showed p-values greater than 0.05.

## Discussion

The objective of this study was to compare the impact of biobanding on training load and technical performance during SSGs in pre-PHV and PHV players from young football squads. The main findings were: 1) The BIO condition differently impacted the PlayerLoad obtained of pre-PHV and PHV players; 2) No effect of the BIO condition was observed for the ITL (session-RPE and TRIMP) and involvement (technical performance during the games) and of pre-PHV and PHV players.

A main novelty of the present study was the higher external training load (average increase of 11.5%) experienced by the pre-PHV players when playing in the bio-banded SSG. Unlike our findings, Arede et al. [22] demonstrated lower PlayerLoad values in pre-PHV basketball players (under-14 to 16) when playing games in the bio-banded compared with the un-matched condition. In addition, our findings differed compared to the results reported by Towlson et al. [13], who showed that external load values of football players were not different

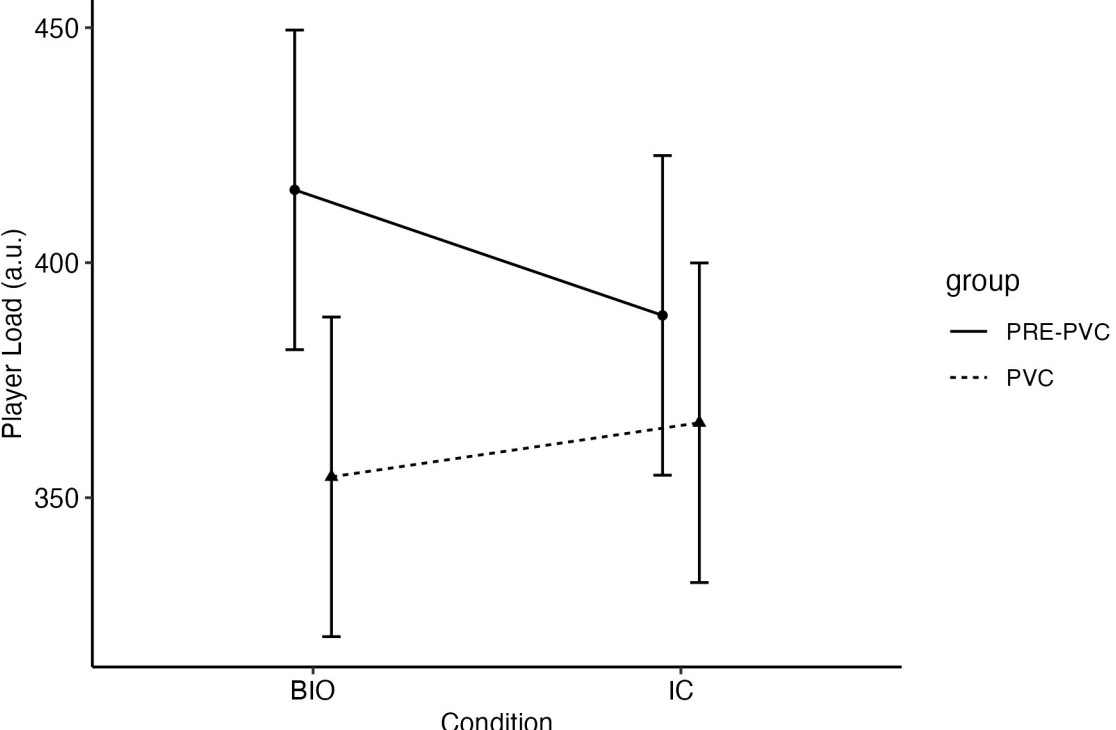

**Fig 1. PlayerLoad for pre-PHV (solid lines) and PHV (dashed lines) players playing in the biobanded and chronological conditions.** Data are estimated means and 95% confidence interval.

when playing SSG in matched and un-matched maturation groups. On the contrary to the external training loads obtained in the pre-PHV players, the PHV group demonstrated higher PlayerLoad values compared to the pre-PHV group during games played at CA. This condition can be partly explained by the age range of the players evaluated, as older players, when playing according to chronological age, were possibly more exposed to the demands of the game compared to the younger athletes. These results are in accordance with Rogol et al. [11], who demonstrated that older football players described the biobanding experience as less physically demanding when compared to traditional age group games. Altogether, our results and others hold significance as conditioning coaches can consider this information to effectively adjust the training workload, particularly when working with groups of athletes approaching the PHV stage.

In the analysis of session RPE, no significant effect of the condition (BIO or CA) was observed. These results differ from the study by [12] that showed a higher subjective measure of exertion in pre-pubescent bio-banded compared to post-PHV players available in SSG. This difference may have been caused by the range of maturation status of our athletes that was not large enough to result in significant differences between the BIO groups. In addition, Abbott et al. [45], reported a game-type effect with the pre-pubescent football players reporting significantly higher session-RPE during the bio-banded competitive games compared to CA games. The difference in the results between these studies can be explained by the competitive stress during the matches, which leads to a greater perception of effort in the athletes [46].

No significant differences were found for TRIMP during the SSG conditions for pre-PHV and PHV players. This result is in line with the values obtained in the monitoring of the

internal training load performed by the RPE [47, 48]. Still, it is worth mentioning that the studies that aimed to verify the HR and TRIMP of athletes grouped by biobanding showed similar results compared to our present investigation. For example, Towlson et al. [13] found no significant differences in mean HR between athletes who performed biobanded SSG. Likewise, Arede et al. [22] showed no differences in TRIMP values in pre-PHV basketball players who performed games in biobanded compared to un-matched games. Important to stress that TRIMP presented very poor reliability across matches played in the BIO and CA conditions.

It is noteworthy that the significant increase found in the ETL in the pre-PHV group may not be relevant in practical application, as this was not able to affect the evaluated internal load parameters. The small difference between the external training load values between BIO and CA groups (~6.5%) might explain why athletes presented similar internal training load (session-RPE) despite significant differences in the external training load during SSG. The lack of differences in the vertical jump between pre- and PHV players may help explaining the lack of difference between groups. It is important to note that the specific age group analysed in this study was intentional, but it should be emphasized that the classification of maturational status was not an inclusion criterion in the present investigation. These findings are in line with the study by Marynowicz et al. [49], which demonstrated in young football players that PlayerLoad values do not correlate with session-RPE. Additionally, a meta-analysis by McLaren et al. [50], showed that the external training load has a trivial to moderate association with training impulse (TRIMP) values.

Regarding technical performance, the reproducibility analyses were low, with a high distribution of zero values. Therefore, all values for each of the technical variables were grouped to determine the technical involvement of the youths in each of the models (CA and BIO). In general, the literature is conflicting, with some studies showing significant effects of biobanding on technical parameters, such as Abbott et al. [45], while others, like Romann et al. [9], have demonstrated no effects. Our study contributes to the literature by also showing no effects of biobanding on technical involvement. For example, on the contrary to Abbott et al. [45] we showed no differences in the amount of dribbling, and in accordance with Romann et al. [9] we also showed no effects of biobanding on technical paraments. The non-significant differences in technical involvement were likely due to the small maturational differences between the groups, as the sample was predominantly composed of prepubescent or pubescent youths. A preferable strategy to increase technical performance during SSG may be altering the SSG rules with an aim to increase contact with the ball and promote a greater player involvement with the actions of the game [28, 51].

As limitations of the study, all athletes were under 13 years-old and the range of maturation status was not large enough to separate between the CA condition. Although previous studies suggest using PHV as a measure for biobanding [6, 52], PHV has a 6-months error which can cause misclassification of the maturation status [53]. However, PHV is an easy-to-apply assessment which increases the external validity of the present investigation. Future studies with larger samples and a wider range of chronological age (for example, up to 15 years of age) and maturational levels are needed. Furthermore, more SSGs across the experimental designs would increase the ability to draw conclusions about the effects of biobanding. A strength of our present investigation is the use of objective assessment of external and internal training load, and the number of SSG performed by the participants.

## Pratical applications

The changes observed in the present investigation suggest that exploring biobanding may be an innovative opportunity for coaches to effectively tailor training sessions, particularly in age

groups below PHV. By grouping athletes based on their biological maturity rather than chronological age, coaches gain a significant tool to better organise training sessions. This approach acknowledges the diverse developmental stages of young athletes, allowing for more targeted and appropriate training, maximizing growth and skill development, especially when there is an early talent promotion process within youth development categories. An important practical application of biobanding is that coaches can conduct maturational band training sessions when they aim to equalize external training loads. An important practical application of biobanding is that coaches can conduct maturational band training sessions when they want to equalize external training loads.

## Conclusion

While no differences were observed for internal training load, biobanding significantly increased external training loads of pre-PHV players compared to the chronological condition and their more mature PHV pairs.

## Author Contributions

**Conceptualization:** Jaimar Fellipe Silva de Macedo, Ricardo Santos Oliveira, Arnaldo Luis Mortatti.

**Data curation:** Jaimar Fellipe Silva de Macedo, Ayrton Bruno de Morais Ferreira.

**Formal analysis:** Jaimar Fellipe Silva de Macedo, Bruno Laerte Lopes Ribeiro, Arnaldo Luis Mortatti.

**Investigation:** Arnaldo Luis Mortatti.

**Methodology:** Jaimar Fellipe Silva de Macedo, Bruno Laerte Lopes Ribeiro, Arnaldo Luis Mortatti.

**Project administration:** Jaimar Fellipe Silva de Macedo, Arnaldo Luis Mortatti.

**Resources:** Ricardo Santos Oliveira.

**Supervision:** Ayrton Bruno de Morais Ferreira, Ricardo Santos Oliveira, Arnaldo Luis Mortatti.

**Writing – original draft:** Jaimar Fellipe Silva de Macedo, Bruno Laerte Lopes Ribeiro.

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
