## [Decision Letter · Decision Letter 0]

7 Jun 2024

PONE-D-24-11339EFFECTS OF BIOBANDING ON TRAINING LOADS AND TECHNICAL PERFORMANCE OF YOUNG FOOTBALL PLAYERSPLOS ONE

Dear Dr. Laerte Lopes Ribeiro,

Thank you for submitting your manuscript to PLOS ONE. After careful consideration, we feel that it has merit but does not fully meet PLOS ONE’s publication criteria as it currently stands. Therefore, we invite you to submit a revised version of the manuscript that addresses the points raised during the review process.

We look forward to receiving your revised manuscript.

Kind regards,

Julio Alejandro Henriques Castro da Costa

Academic Editor

PLOS ONE

Journal Requirements:

Reviewers' comments:

Reviewer's Responses to Questions

**Comments to the Author**

1. Is the manuscript technically sound, and do the data support the conclusions?

Reviewer #1: No

Reviewer #2: Partly

2. Has the statistical analysis been performed appropriately and rigorously? 

Reviewer #1: Yes

Reviewer #2: No

3. Have the authors made all data underlying the findings in their manuscript fully available?

Reviewer #1: No

Reviewer #2: Yes

4. Is the manuscript presented in an intelligible fashion and written in standard English?

Reviewer #1: Yes

Reviewer #2: Yes

5. Review Comments to the Author

**Reviewer #1:** Dear authors, we have found that there may be the following problems in your paper, which need your explanation or appropriate modification:

1. Please explain how subjects are recruited and screened.

2. Lower extremity strength usually increases with the age of adolescents, and the CMJ of adolescent athletes who are older and participate in sports for a longer time is usually higher than that of younger athletes. Please explain why the older PHV group in your study had significantly lower CMJ than the pre-PHV group of athletes. If possible, please provide raw data of the subject's Age, yPHV, Height, Body mass, CMJ, as these will affect their internal and external load and SSGs performance, which is one of the factors that must be considered for grouping.

3. Data may be wrong. In Table 2, the age, height, weight and other data of athletes in PHV group and pre-PHV group are different from the mean value of all athletes after calculation. For example, adding the CMJ data for athletes in the PHV and pre-PHV groups yielded a CMJ mean (25.315) that was different from the CMJ mean for all athletes (25.03).

4. In the table with significant differences between the two groups of data, please report the specific P-value and other relevant index values.

5. Since all the athletes in your study were under the age of 13, the differences in maturity were not significant enough. Differences in body weight, training level, and field position of the subjects in the study can also interfere with the results, and it should be explained how to control for these variables.

**Reviewer #2:** I would like to thank the authors for this study examining an interesting and relevant issue regarding maturational differences and the effects of biobanding on technical performance. Please find my comments below.

1. One of my major concerns is the presentation of the research data. More specifically, in addition to the results presented in the manuscript, it would be useful to see also the results for each session separately. By doing this, a clearer view on data variability and consistency according to condition (BIO or CA) could be examined.

2. Also, a clear explanation is needed on the use of the external load (as applied here) in the examined age range. Does it really reflect true load in such young age? Why changes in acceleration was preferred over other metrics (e.g. total distance, running speed)? And how is it connected to the internal load (HR)? Please provide a strong theoretical background supported by appropriate references.

3. The study includes players ranging from 10 to 13 years. How reasonable is to have children of 3 years difference in CA grouped together? Please, justify the selection of this age range.

Abstract

Please include a short description on the rationale of the study. EM Abbreviation is not explained in the abstract. The conclusions seem to repeat the results, please provide conclusions and the practical relevance of the results in the last paragraph.

Introduction

In general, introduction is well-written. Again, a justification of the indicators used to assess external load in small- sided games would contribute to better understand the results. In its present form it is not convincing that what we measure is what we want to know.

When using in-text citations please reconsider according to: Author et al. (1) found no effects on…

Ln 68. This sentence seems not to be completed: reduce discrepancies caused by…???

Ln 76-78. This sentence is not clear, please consider rephrasing. What do you mean by morpho functional adaptations?

Ln 78. Consider starting a new sentence here (In this way,….)

Methods

It is not clear how grouping based on CA was performed (detailed explanation is needed). Again, I don’t think that is usual to group boys in this age range (10 to 13 years) together.

What maturational differences we may expect in 10-year-old boys? It is easy to think that maturation does not affect performance, considering that at this age boys are still well before their APHV. Otherwise, please explain providing convincing references.

Data of the different conditions were summed/averaged per game? Please, include this information in the manuscript.

Ln 190. A reference is needed here.

Considering the low sample size effect size results seem necessary to confirm any meaningful differences. Please add to the statistical analysis a description of the inter-session reliability (ICC calculation), which is reported in the results.

Results

Differences in maturational status may be also explained with the differences in their CA. PHV players in PHV group were 1.5 year older than those in the pre-PHV group. In this sense we cannot speak about players of the same CA, but of different maturational status.

ICC values seem too low (e.g. for TRIMP, technical parameters). I don’t think they could be averaged in this way.

CMJ did not differ between PHV groups (even more, pre-PHV players achieved higher performance than PHV players) indicating that maturation had no effects on performance. Why then we should expect differences in technical variables or player load?

Ln244-247. Description of the results seem confusing. In line245 the increase of 11.5% in CA condition compared to BIO condition refer to the pre-PHV group (as stated in the manuscript) or to the PHV group? (as illustrated in figure 1). Please clarify and correct accordingly.

Ln 253-254. Similarly to previous comment, please check.

Discussion

Discussion is well written and easy to follow. What may be a possible explanation for the increase in external load for the pre-PHV group under biobanding grouping? Biobanding however, seems less effective for the players in PHV group, this result is not discussed in the manuscript, and I think it deserves more attention.

Ln. 294. Please check for grammar and consider revising (…PlayerLoad obtained of pre-PHV….)

Ln315-316- Please check for grammar and consider revising.

Ln340. Format of the in-text citation!!! (Romann et al., 2014).

Ln356. Exactly, this is what I meant earlier, all players were under 13 years, having therefore few (if any) differences in maturation. The design here included players of differences in CA, but not definitely in maturation. I think a reverse design would be more useful, that is players of the same CA, but of distinct maturational status.

References are not formatted according to the journal’s instructions. Please check and correct accordingly.

Please mark significant differences in figure 1.

6. PLOS authors have the option to publish the peer review history of their article (what does this mean?). If published, this will include your full peer review and any attached files.

Reviewer #1: No

Reviewer #2: No

---

## [Author Response · Author response to Decision Letter 0]

26 Jul 2024

July 26, 2024

RE: Revised version of the manuscript - PONE-D-24-11339 “EFFECTS OF BIOBANDING ON TRAINING LOADS AND TECHNICAL PERFORMANCE OF YOUNG FOOTBALL PLAYERS”.

PLOS ONE”

Dear Editor-in-Chief,

First of all, we appreciate the time devoted to evaluating our manuscript, helping us to craft hopefully an improved version. We are pleased to clarify your concerns, which we believe will improve the impact and quality of our work. We have made a concerted attempt to systematically address the specific concerns raised for this revision. We have highlighted the alterations in the manuscript in yellow for convenience. Your constructive and insightful points have improved selected areas of the manuscript significantly. We believe that the manuscript has been significantly improved based on your insightful comments and we hope that based on these modifications, the manuscript is now closer to acceptance.

Reviewer Comments:

Reviewer #1: Dear authors, we have found that there may be the following problems in your paper, which need your explanation or appropriate modification:

1. Please explain how subjects are recruited and screened.

RESPONSE: Thank you for this comment. The young athletes were recruited based on convenience, as they were already participating in the youth divisions of the club where the research was conducted. This approach is justified by the nature of the study, which required that all young athletes meet the primary inclusion criterion of being trained under the same conditions, i.e., the same training duration, number of weekly training sessions, and external training load. This information was included in the text as follows:

 Twenty young footballers (11.8 ± 1.15 years), volunteered for this study. They were recruited based on convenience to meet the inclusion criteria of the study: (A) participation in the under-13 category, (B) training with the group for at least six months, and (C) having attended at least 80% of the training sessions during this period.

2. Lower extremity strength usually increases with the age of adolescents, and the CMJ of adolescent athletes who are older and participate in sports for a longer time is usually higher than that of younger athletes. Please explain why the older PHV group in your study had significantly lower CMJ than the pre-PHV group of athletes. 

RESPONSE: We appreciate the reviewer's comments. Please note that CMJ differences were not significant between groups (p=0.420). This condition is supported by the studies of Jones et al. (2020) that showed no significant difference for the Eccentric knee flexor in Under-11, 12 and 13 groups and Moreira et al. (2013) which also did not indicate significant differences between groups that had not surpassed PHV. 

2.1 If possible, please provide raw data of the subject's Age, yPHV, Height, Body mass, CMJ, as these will affect their internal and external load and SSGs performance, which is one of the factors that must be considered for grouping.

RESPONSE: Thank you for this comment. The aim of our study was to investigate weather grouping young players according to somatic maturation would lead to a better training load profile during small-sided games. Therefore, as per experimental manipulation differences in maturational levels were significant between groups (-2.2 yPHV vs. -0.89 yPHV). Furthermore, the method used to determine maturational considered weight, height, and age of the participants. As such, we understand that considering all variables will result in excessive control of variables and will mischaracterize the biobanding strategy under scrutiny in our study.

3. Data may be wrong. In Table 2, the age, height, weight and other data of athletes in PHV group and pre-PHV group are different from the mean value of all athletes after calculation. For example, adding the CMJ data for athletes in the PHV and pre-PHV groups yielded a CMJ mean (25.315) that was different from the CMJ mean for all athletes (25.03).

RESPONSE: We appreciate the careful review of our study. However, the presented data is not wrong. The average of the whole group for CMJ was indeed 25.03. If you take the average of two subgroups and calculate an average of them both the value will be slightly different than the whole group average. 

4. In the table with significant differences between the two groups of data, please report the specific P-value and other relevant index values.

RESPONSE: We appreciate the reviewer's comments. Please see the text for exact p-values. Inserting them again in the tables will lead to unnecessary duplication of information.

5. Since all the athletes in your study were under the age of 13, the differences in maturity were not significant enough. Differences in body weight, training level, and field position of the subjects in the study can also interfere with the results, and it should be explained how to control for these variables.

RESPONSE: Thank you for this comment. Firstly, as per experimental manipulation we stress that the maturation level between groups was significant. The PHV group were at PHV with an average maturation of -0.89 ± 0.36 yPHV and the pre-PHV were at -2.2 ± 0.67 yPHV. These groups were significantly different regarding their maturation levels. A possible limitation of our study is the lack of a post-PHV group. This information has been discussed in our submitted article (please see page 12)

As discussed in item 2.2 of this review, we understand that body weight is impacted by maturation and that is the reason why we conducted the study. Further control for this variable is beyond the scope of our article. 

Moreover, training level is an important consideration and we tried to minimize by selecting a group of participants, who albeit of different ages, were participating in the same type of training for at least 6 months.

Regarding players positions, participants did not have clearly defined roles in the squad. This information was inserted in the text as follow:

 Considering the category and the age groups included the positions on the field have not yet been clearly defined.

Reviewer #2: I would like to thank the authors for this study examining an interesting and relevant issue regarding maturational differences and the effects of biobanding on technical performance. Please find my comments below.

RESPONSE: Thank you for your kind comment. 

6. One of my major concerns is the presentation of the research data. More specifically, in addition to the results presented in the manuscript, it would be useful to see also the results for each session separately. By doing this, a clearer view on data variability and consistency according to condition (BIO or CA) could be examined.

RESPONSE: We appreciate the reviewer’s comments. Using two SSGs in each condition is a strength of our investigation. This is because playing twice in the same condition would increase certainty in the observed findings. Furthermore, to model the impact of the condition, maturation and interaction we used a linear mixed-model approach with a repeated design to include both matches with each participant with 4 observations in the model (80 observation). From there, we inserted random effects for participants. We are confident that this approach is the best to deal with the repeated design rather than averaging the two matches into one score. Presenting both sessions separately rather than modeling the date would lead to uncertainty.

7. Also, a clear explanation is needed on the use of the external load (as applied here) in the examined age range. Does it really reflect true load at such young age? Why changes in acceleration were preferred over other metrics (e.g. total distance, running speed)? And how is it connected to the internal load (HR)? Please provide a strong theoretical background supported by appropriate references.

RESPONSE: Thank you for this suggestion. PlayerLoad was used as a measure of external load because it is a more comprehensive metric, quantifying the total movement performed by players across three axes (horizontal, vertical, and lateral). Other studies have used the same method with athletes of similar age groups (Castellano et al, 2016 and Silva et al, 2024 ) and with young athletes [17 ± 0.9 years (Denmark et al, 2021 )] demonstrating its effectiveness in this population. Additionally, PlayerLoad is an indicator that correlates with the session RPE method [(r = .51, p < .001) (Silva et al, 20244)] and Edwar’s method (Askow et al, 2021 ) (Casamichana et al, 2012 ). This information has been included in the manuscript, please see page 06.

8. The study includes players ranging from 10 to 13 years. How reasonable is to have children of 3 years difference in CA grouped together? Please, justify the selection of this age range.

RESPONSE: Thank you for this observation. Our investigation was conducted inside a football club where children unde-13 trained together. This included children aged 10-13 years old. This age range in training and competition is common in our country, especially in football. The age range inside the under-13 category is one of the rationales why we believe biobanding should be used for training. 

Abstract

9. Please include a short description on the rationale of the study. EM Abbreviation is not explained in the abstract. The conclusions seem to repeat the results, please provide conclusions and the practical relevance of the results in the last paragraph.

RESPONSE: Thank you for this suggestion. We have now inserted the short rationale in the abstract as follow:

 Background: In adolescent sports, grouping by chronological age can advantage athletes born early in the year due to maturation differences. Early maturing athletes often achieve greater performance gains, are perceived as more talented, and receive more specialized training and workloads. This study aimed to assess the effects of biobanding on internal and external training loads, as well as technical performance during small-sided games (SSGs).

Introduction

10. In general, introduction is well-written. Again, a justification of the indicators used to assess external load in small- sided games would contribute to better understand the results. In its present form it is not convincing that what we measure is what we want to know.

RESPONSE: Thank you for this suggestion. We have now inserted a paragraph in the introduction as follow:

 Thus, using external training load analysis instruments, such as the PlayerLoad method—which quantifies total movement across three axes (horizontal, vertical, and lateral)—along with measuring session-RPE and training impulse (TRIMP), enables the assessment of internal training load using both objective and subjective data (Casamichana et al. 2012). This approach allows for a comprehensive analysis of the impact of external load on the player. By combining external load values with internal load measurements, we can evaluate a player's ability to handle the training session (Bourdon et al. 2017). On the other hand, analyzing technical parameters of the game can provide valuable insights into a player's involvement, depending on their maturational status. Biobanding, in particular, can create more opportunities for late-maturing players to utilize their technical competence effectively. (Lüdin et al. 2022).

11. When using in-text citations please reconsider according to: Author et al. (1) found no effects on…

RESPONSE: Thank you for your observation. Done accordingly.

12. Ln 68. This sentence seems not to be completed: reduce discrepancies caused by…???

RESPONSE: Thank you for your observation. This phrase has been removed from the text.

13. Ln 76-78. This sentence is not clear, please consider rephrasing. What do you mean by morpho functional adaptations?

RESPONSE: Thank you for this comment. The sentence has been rewritten as follow:

 In the context of football training, using methods that facilitate both morphological and functional adaptations and the acquisition of sport-specific skills may prove superior, especially in early years of athletic development

14. Ln 78. Consider starting a new sentence here (In this way,…)

RESPONSE: Done accordingly.

Methods

15. It is not clear how grouping based on CA was performed (detailed explanation is needed). Again, I don’t think that is usual to group boys in this age range (10 to 13 years) together. What maturational differences we may expect in 10-year-old boys? It is easy to think that maturation does not affect performance, considering that at this age boys are still well before their APHV. Otherwise, please explain providing convincing references.

RESPONSE: Thank you for this comment. Regarding the how grouping based on CA was performed, the explanation is inserted in the text as follow:

 1) CA - SSGs with players divided according to their chronological age (i.e., Randomized teams without the use of biobanding). The CA group was randomized while ensuring that all age ranges were similarly represented in each of the 4 teams formed. Each team included players aged 10, 11, 12 and 13 years old, ensuring balanced age distribution across all teams.

Furthermore, the authors understand the reviewer's concern; however, as explained in item 8, only one player was younger than those normally seen in the under-13 category. In the authors' opinion, the possible maturational difference of this athlete could increase the likelihood of differences in results between the groups (BIO and CA). However, since the athlete played in both conditions, the possible differences (chronological and biological) would be equally distributed between the groups, which would not interfere with the final result. Analyses without this athlete produced results very similar to those presented in the study.

The authors agree with the reviewer that maturational status does not affect performance in the years before PHV , , likewise, the majority of the results from this research support this statement. Most of the athletes had not reached or surpassed PHV, and this was decisive for the final research result.

16. Data of the different conditions were summed/averaged per game? Please, include this information in the manuscript.

RESPONSE: Thanks for this comment. Our statistical model handles repeated data in the models. As such, we inserted both games in the models and adjusted per participant accordingly. Please see the statistical section for details. 

17. Ln 190. A reference is needed here.

RESPONSE: Thanks for this comment. Done accordingly (DOI: 10.1039/C5JA00380F)

18. Considering the low sample size effect size results seem necessary to confirm any meaningful differences. Please add to the statistical analysis a description of the inter-session reliability (ICC calculation), which is reported in the results.

RESPONSE: We have presented the estimated effects from the linear mixed-models. We are unaware of any other effect size estimate for the statistical approached used in our study. If the reviewer has a different understanding of effect and could point us to where we could find reliable effect sizes for linear mixed-models we would appreciate. However, please note that the presented data in Table 4 are estimated effects (and associated uncertainty) from the linear mixed models. These can be used as effects estimates. For example, it was observed that for external training load the estimated effect for the chronological condition was -26.71 AU compared to BIO. 

Results

19. Differences in maturational status may be also explained with the differences in their CA. PHV players in PHV group were 1.5 year older than those in the pre-PHV group. In this sense we cannot speak a bout player of the same CA, but of different maturational status. 

RESPONSE: Thanks for this comment. In fact, the reviewer is correct; however, we did not specifically address players of the same CA. the CA group refers to the under-13 category (ranged 10-13 years old). This information was insert in the Design section as follow:

This experimental study involved participants performing SSGs under two different conditions: 1) CA - SSGs with players divided according to their chronological age (i.e., Randomized teams without the use

---

## [Decision Letter · Decision Letter 1]

16 Aug 2024

PONE-D-24-11339R1EFFECTS OF BIOBANDING ON TRAINING LOADS AND TECHNICAL PERFORMANCE OF YOUNG FOOTBALL PLAYERSPLOS ONE

Dear Dr. Laerte Lopes Ribeiro,

Thank you for submitting your manuscript to PLOS ONE. After careful consideration, we feel that it has merit but does not fully meet PLOS ONE’s publication criteria as it currently stands. Therefore, we invite you to submit a revised version of the manuscript that addresses the points raised during the review process.

We look forward to receiving your revised manuscript.

Kind regards,

Julio Alejandro Henriques Castro da Costa

Academic Editor

PLOS ONE

Reviewers' comments:

Reviewer's Responses to Questions

**Comments to the Author**

1. If the authors have adequately addressed your comments raised in a previous round of review and you feel that this manuscript is now acceptable for publication, you may indicate that here to bypass the “Comments to the Author” section, enter your conflict of interest statement in the “Confidential to Editor” section, and submit your "Accept" recommendation.

Reviewer #1: (No Response)

Reviewer #2: All comments have been addressed

2. Is the manuscript technically sound, and do the data support the conclusions?

Reviewer #1: No

Reviewer #2: Yes

3. Has the statistical analysis been performed appropriately and rigorously? 

Reviewer #1: No

Reviewer #2: Yes

4. Have the authors made all data underlying the findings in their manuscript fully available?

Reviewer #1: No

Reviewer #2: Yes

5. Is the manuscript presented in an intelligible fashion and written in standard English?

Reviewer #1: Yes

Reviewer #2: Yes

6. Review Comments to the Author

**Reviewer #1: **1. Before recruiting participants, did the authors calculate the minimum sample size required for the study? Please report the specific calculation method and results.

2. In the Statistical analysis section, did the authors analyze the independence, distribution, and variance of all indicator data before using the T-test? It is necessary to report whether the sample data meet the assumptions required for the use of the independent samples T-test and Mixed model linear regression methods before using them. If some indicators in the study do not apply to the above methods (such as non-linearity, not meeting the normal distribution), it will lead to unreliable research results and conclusions, and other methods should be considered for analysis. It may be inappropriate to present data in the form of mean and standard deviation before knowing the distribution of the data. From the data submitted by the authors, some data seem not to conform to the normal distribution and homogeneity of variance assumptions.

3.Are the ICC values reported in lines 231-233 the ICC values between the individual internal tests of the participants? If possible, please report the ICC and CV values for all test indicators to enhance the reliability of the results. In addition, the ICC values for RPE, TRIMP, and technical variables (each indicator's ICC should be reported specifically) are below the generally acceptable minimum standard in research (ICC > 0.75 and CV < 15%) [1-4]. From the submitted data, it appears that there is a large difference in data for the same athlete under the same conditions in two competitions.

[1]Baumgartner TA, Jackson AS. Measurement for evaluation in physical education and exercise science.Meas Eval Phys Educ Exerc Sci. 1998. https://doi.org/10.4324/9781315312736.

[2]Baumgartner TA, Chung H. Confidence limits for intraclass reliability coefficients. Meas Phys Educ Exerc Sci 5: 179–188, 2001.

[3]Safrit, M. J., & Wood, T. M. (1995). Introduction to measurement in physical education and exercise science (3rd ed.). New York: Mosby.

[4]Atkinson G, Nevill AM. Statistical methods for assessing measurement error (reliability) in variables relevant to sports medicine. Sports Med. 1998;26(4):217-238. doi:10.2165/00007256-199826040-00002

4. Should Table 1 on line 228 be Table 2?

5. The average value of each group in Table 2, multiplied by the number of people in each group and then added together, divided by the total number of people to get the average value (such as CMJ), has a significant difference from the overall average value in Table 2. This difference is greater than the general error of data processing, please ask the authors to confirm whether there may be calculation errors in the data. In addition, the data involved in this study (such as the original data of Age, yPHV, Height, Body mass, CMJ, etc., in Table 2) have not all been uploaded to the database, and the above data should be displayed in the database as part of the study.

6. In the data uploaded by the authors, it was found that many athletes' data are not completely complete, such as in the data of steal_complete, steal_incomplete, tackle_incomplete, and other indicators, many athletes do not have data points, the lack of the above data points will affect the statistical power and significance, and to some extent affect the results and conclusions. Therefore, the actual number of data points for each indicator should be reported in the table (such as adding the number of data points for each indicator in Table 3). In addition, according to the general reporting standards, should the data such as Total shots (n) in Table 3 be reported to one decimal place like other data?

7. The specific statistical power, fixed-effects, and random effects test results should be reported in Tables 4 and 5 (or in the text).

8. I believe that the number of tests conducted in your study is relatively small, and there is a large difference in the performance of the same subjects in different training and competitions within your research. Therefore, the results of the two tests may not adequately represent the true athletic performance of the athletes. Additionally, factors such as the physical condition of the athletes, their relative positions on the field, and the level of defense against the subjects may vary across different tests, and these factors can significantly affect the subjects' performance. The authors did not control for these factors in their study, so the results may not support the authors' conclusions.It is recommended to increase the number of tests to better understand the true conditions of the subjects, and to control the impact of the aforementioned factors on the results.

**Reviewer #2:** I would like to thank the authors for their revision considering almost all previous comments. A convincing explanation of the age range examined in this study seems to be still missing, however, this does not significantly affect the manuscript's quality. The common practice to group athletes of 10-13 years in one age-group (as stated by the authors) seems to induce unbalanced training loading, based on the results the authors could consider including clear recommendations for policy makers and club administration for a more effective grouping method.

7. PLOS authors have the option to publish the peer review history of their article (what does this mean?). If published, this will include your full peer review and any attached files.

Reviewer #1: No

Reviewer #2: No

---

## [Author Response · Author response to Decision Letter 1]

13 Dec 2024

October 29, 2024

RE: Revised version of the manuscript - PONE-D-24-11339 “EFFECTS OF BIOBANDING ON TRAINING LOADS AND TECHNICAL PERFORMANCE OF YOUNG FOOTBALL PLAYERS”.

PLOS ONE”

Dear Editor-in-Chief,

First of all, we appreciate the time devoted to evaluating our manuscript, helping us to craft hopefully an improved version. We are pleased to clarify your concerns, which we believe will improve the impact and quality of our work. We have made a concerted attempt to systematically address the specific concerns raised for this revision. We have highlighted the alterations in the manuscript using the track changes tool. Your constructive and insightful points have improved selected areas of the manuscript significantly. We believe that the manuscript has been significantly improved based on your insightful comments and we hope that based on these modifications, the manuscript is now closer to acceptance.

Reviewer Comments:

Reviewer #1: 

1. Before recruiting participants, did the authors calculate the minimum sample size required for the study? Please report the specific calculation method and results.

RESPONSE: No a priori sample size calculation was done. We investigated the sport club we had access to, in which there were 20 children playing in the team. We proposed the research for the team staff, they agreed to participate, and all children were involved. Unfortunately, having a limited number of participants is a limitation when working in a real-life setting inside a sports club. 

Furthermore, it is worth highlighting that performing power calculation for linear mixed model (LMM) is not a trivial task (please see: https://doi.org/10.3758/s13428-021-01546-0). The choice for a LMM approach is that this analysis is able to model the structure of the data (repeated measures) and provide estimates considering variability within/between subjects. 20 participants with four observations each allows the model to tease apart the effects of the experimental design on our dependent variables. 

If this reviewer is aware of any a posteriori sample size calculations and, most importantly, how the 20 participants in our sample limits the power in our analysis, we are happy to follow and proceed accordingly. 

In the manuscript we have acknowledged in the limitation that the results are limited to a small sample size with future larger investigations needed to strengthen the biobanding hypothesis. 

2. In the Statistical analysis section, did the authors analyze the independence, distribution, and variance of all indicator data before using the T-test? It is necessary to report whether the sample data meet the assumptions required for the use of the independent samples T-test and Mixed model linear regression methods before using them. If some indicators in the study do not apply to the above methods (such as non-linearity, not meeting the normal distribution), it will lead to unreliable research results and conclusions, and other methods should be considered for analysis. It may be inappropriate to present data in the form of mean and standard deviation before knowing the distribution of the data. From the data submitted by the authors, some data seem not to conform to the normal distribution and homogeneity of variance assumptions.

RESPONSE: We thank the reviewer for the careful assessment of our article. For the comparisons between groups presented in Table 2, we have now checked for normality of distribution and homogeneity of variances. In cases when not met, we applied non-parametric tests (please see our answer to question #5 below). It is worth noting that non-parametric statistics let to virtually exact the same results and interpretation. 

For the LLMs (and GLLMs see below). Our data structure contains dependent data points (e.g., same participant observed in different moments) and the assumption of independence is not met. Hence why we have used linear mixed models, which accounts for the repeated nature of our investigation (e.g., same participants with 4 data points in all analysis). Moreover, given the predictors (independent variables) are binary (either one group/condition or the other) the coefficients will always meet the assumption of linearity.

Regarding the normal distribution, it is paramount to stress that for (G)LLMs the distribution of the dependent variable and homogeneity of variances are not assumptions to be met (please see relevant chapters in: 1 -"Discovering statistics Using r by Andy Field, Jeremy Miles, and Zoë Field; 2012" for a very basic introduction; and 2 - "Linear Mixed-Effects Models Using R. A Step-by-Step by Approach by Andrzej Gałecki and Tomasz Burzykowski" for a more detailed discussion). 

However, as per reviewer's suggestion, for the LLMs we have also assessed normality for the training load variables using the Shapiro wilk test for Session-RPE (P = 0.002), player load total (P = 0.15) and TRIMP (P = 0.14). As observed, Session-RPE did not follow a normal distribution which did not alter the normal distribution of the residuals. Please see below visual analysis and Shapiro Wilk's results in parenthesis. 

Trimp residual distribution (W = 0.98061, p-value = 0.2656).

Session-RPE residual distribution (W = 0.98622, p-value = 0.5486).

Player load total (W = 0.97953, p-value = 0.2281).

Regarding the homogeneity of the variances (heteroscedasticity). Although not an assumption, this was checked looking at the fitted vs the predicted plots from the regression. Please see below for the models of training load: 

Trimp model: 

Session-RPE model: 

Player load model: 

Although a tendency seems to exist for TRIMP, this is not affecting the models. To test that, we inserted random effects for the intercept and slopes (to model the possible impact of heteroscedasticity). All model statistics (bayes factor, AIC, BIC, and p values) favor the model with only random intercepts, and models led to the same predictions (please also note the linearity) as follows:

Session-RPE 

Playerload

TRIMP (more about TRIMP on answer to #3)

Data in figures are predictions of the dependent variable by condition (CA and BIO) split by maturation status. Red line: model with random intercepts and slopes; Green line: model with random intercepts only. 

Finally, technical performance variables used in the investigation (total of passes, shots, etc). These variables are originally summed count data (e.g., number of complete passes summed by the number of incomplete passes). Importantly, technical actions do not follow a normal distribution, but rather a Poisson distribution, in some cases with the presence of zero inflated values:

Total tackle:

Total steals:

Total Shots:

In addition to the 0-inflated characteristic, we also agree with the reviewer (answer #3) that reliability of the technical actions is an issue. Furthermore, given the open nature of the SSGs reliability of all technical actions were very low (please see our answer to question #3). For this, in accordance with the reviewer, we decided not to test the individual technical variables and rather decided to only analyze involvement (sum of every technical action in the SSGs) the data. 

Involvement was analyzed using a Linear Mixed Models, which failed to meet the assumptions after being log transformed:

Involvement:

Residuals involvement (W = 0.9702, p-value = 0.05848)

Due to this, we have also analyzed involvement with a mixed model generalized linear model (GLMM) with a Poisson distribution: 

Left linear mixed model/right Poisson GLMM

 Involvement

Predictors Estimates CI p Incidence Rate Ratios CI p

(Intercept) 3.18 2.96 – 3.41 <0.001 25.46 20.55 – 31.53 <0.001

cond [IC] -0.15 -0.46 – 0.16 0.350 0.89 0.66 – 1.20 0.442

st [PVC] -0.01 -0.33 – 0.31 0.952 1.00 0.74 – 1.35 0.994

cond [IC] × st [PVC] 0.20 -0.24 – 0.63 0.379 1.18 0.77 – 1.81 0.445

Random Effects

σ2 0.24 0.04

τ00 0.01 id 0.10 id

τ11 0.20 id.condIC

ρ01 -0.70 id

ICC 0.04 0.72

N 20 id 20 id

Observations 80 80

Marginal R2 / Conditional R2 0.019 / 0.058 0.027 / 0.725

Looking at the table above at the Poisson model, contains a random slope and the model was favored by the AIC and BIC statistics. 

Given all the above, and considering the reviewer’s suggestions, in the new version: 

Table 2 has been adjusted to include comparisons using non-parametric Mann Whitney test in cases when normality of distribution was not met. Furthermore, data in table was triple checked and now contains the right values (please see our answer to your question #5).

We have deleted the technical performances from the analysis and only included involvement and a variable to indicate overall participation in the SSGs matches. Also, involvement was analyzed using a Poisson GLLMs with a random intercept for participants to account for the repeated (independence) nature of our dataset and a random slope which improved significantly the prediction of the model. This information has been included in the statistical analysis section.

We have also included in the discussion a paragraph discussing the poor reliability of the technical data and how this may impact interpretation (please see our answer to #3). 

3. Are the ICC values reported in lines 231-233 the ICC values between the individual internal tests of the participants? If possible, please report the ICC and CV values for all test indicators to enhance the reliability of the results. In addition, the ICC values for session-RPE, TRIMP, and technical variables (each indicator's ICC should be reported specifically) are below the generally acceptable minimum standard in research (ICC > 0.75 and CV < 15%) [1-4]. From the submitted data, it appears that there is a large difference in data for the same athlete under the same conditions in two competitions.

[1]Baumgartner TA, Jackson AS. Measurement for evaluation in physical education and exercise science.Meas Eval Phys Educ Exerc Sci. 1998. https://doi.org/10.4324/9781315312736.

[2]Baumgartner TA, Chung H. Confidence limits for intraclass reliability coefficients. Meas Phys Educ Exerc Sci 5: 179–188, 2001.

[3]Safrit, M. J., & Wood, T. M. (1995). Introduction to measurement in physical education and exercise science (3rd ed.). New York: Mosby.

[4]Atkinson G, Nevill AM. Statistical methods for assessing measurement error (reliability) in variables relevant to sports medicine. Sports Med. 1998;26(4):217-238. doi:10.2165/00007256-199826040-00002

RESPONSE: We thank the reviewer for the comments about reliability. However, the aim of our investigation is not to present the reliability of technical and training load variables during small-sided games (SSG). This is expected to occur given the open nature of training routines, specially SSGs. With the low reliability (high variability in the technical and training load performance during SSGs) in mind, we decided to use two training sessions per experimental manipulations. This was done to try and get closest to the true technical and physiological responses of the SSGs in children, and two similar SSG is also what is realistic for an experimental manipulation inside the training routine of the team.

However, we have now inserted matches (SSG #1 and SSG #2) as a fixed effect variable to test whether players changed the outcomes across matches. Random slopes were tested but they did not increase the model fit. This was done to check for changes in the outcome variable across the two repeated SSGs as one indicative of reliability (we did not expect the variables to change from SSGs to SSGs). 

The table below describes the effect of SSG matches on the outcome of the study. 

Table reliability of the variables included in the manuscript. Models are LMMs 

 TRIMP PLOT Session-RPE

Predictors Estimates CI p Estimates CI p Estimates CI p

Intercept 107.88 101.95 – 113.81 <0.001 394.22 366.41 – 422.03 <0.001 5.35 4.63 – 6.07 <0.001

jog2 -15.72 -23.– -8.12 <0.001 -18.50 -41.02 – 4.01 0.106 0.75 -0.04 – 1.54 0.061

condIC -15.63 -23.22 – -8.03 <0.001 -11.63 -34.15 – 10.88 0.307 0.05 -0.74 – 0.84 0.899

jog2:condIC 22.06 11.31 – 32.80 <0.001 8.07 -23.78 – 39.91 0.615 -0.70 -1.81 – 0.41 0.213

Random Effects

σ2 145.46 1276.91 1.55

τ00 31.52 id 2618.92 id 1.03 id

τ11 

ρ01 

ICC 0.18 0.67 0.40

N 20 id 20 id 20 id

Observations 80 80 80

Marginal R2 / Conditional R2 0.191 / 0.335 0.018 / 0.678 0.035 / 0.419

log-Likelihood -308.997 -406.701 -142.854

Reliability of the count technical variables included in the manuscript. Models are Poisson with a random intercept and a random slope

As clearly observed, the variable TRIMP changed across the SSGs with an interaction between condition and TRIMP. 

Because of the changes observed for the TRIMP variable across matches, we have now adjusted the effects of condition for matches. This did not change the observed results. 

We have now discussed the poor reliability is in the new version of the manuscript and inserted a paragraph with the findings of the changes in SSG #1 compared to SSG #2 (e.g., reliability). Importantly, the ICC used in the analysis and article was obtained from the (G)LMMs. This is stronger than using separated analysis as (G)LMMs consider the variability between conditions as well as within participants whilst accounting for the dependence in the observations. 

Please see the results discussion section where we present the ICC values from the linear-mixed models and address the issue of poor reliability of the performance variables during SSGs. 

4. Should Table 1 on line 228 be Table 2?

RESPONSE: Done accordingly. 

5. The average value of each group in Table 2, multiplied by the number of people in each group and then added together, divided by the total number of people to get the average value (such as CMJ), has a significant difference from the overall average value in Table 2. This difference is greater than the general error of data processing, please ask the authors to confirm whether there may be calculation errors in the data. In addition, the data involved in this study (such as the original data of Age, yPHV, Height, Body mass, CMJ, etc., in Table 2) have not all been uploaded to the database, and the above data should be displayed in the database as part of the study.

RESPONSE: We have now carefully inspected the results of Table 2. We found the mistake as in Brazil commas are used to separate decimal places and when moving to the statistical software used to describe participants a mix of values separated either by commas or dots led to different descriptive averages. We have now reassessed the data and tripled checked all values presented. We can confirm the data now is correct and we have also uploaded it to the repository. 

Furthermore, data in Table 2 was also assessed for normality (Shapiro Wilk) and homogeneity of variances (Levene test). In cases when normality of distribution (age, body mass, and cmj) and homogeneity of variances (age and body mass) was an issue, we used the Mann Whitney Rank test to compare the descriptive variables. This information has now been added to the manuscript and Table 2 corrected accordingly.

6. In the data uploaded by the authors, it was found that many athletes' data are not completely complete, such as in the data of steal_complete, steal_incomplete, tackle_incomplete, and other indicators, many athletes do not have data points, the lack of the above data points will affect the statistical power and significance, and to some extent affect the results and conclusions. Therefore, the actual number of data points for each indicator should be reported in the table (such as adding the number of data points for each indicator in Table 3). In addition, according to the general reporting standards, should the data such as Total shots (n) in Table 3 be reported to one decimal place like other data?

RESPONSE: We have adjusted the lack of data with zeros. Those missing values were there because in some situations athletes did not perform the technical parameter under assessment. We have now replaced missing by zeros and all analysis about technical performance were completed once again. Please see our answer to your question #2. 

7. The specific statistical

---

## [Decision Letter · Decision Letter 2]

30 Dec 2024

EFFECTS OF BIOBANDING ON TRAINING LOADS AND TECHNICAL PERFORMANCE OF YOUNG FOOTBALL PLAYERS

PONE-D-24-11339R2

Dear Dr. Laerte Lopes Ribeiro,

We’re pleased to inform you that your manuscript has been judged scientifically suitable for publication and will be formally accepted for publication once it meets all outstanding technical requirements.

Kind regards,

Julio Alejandro Henriques Castro da Costa

Academic Editor

PLOS ONE

Additional Editor Comments (optional):

Reviewers' comments:

Reviewer's Responses to Questions

**Comments to the Author**

1. If the authors have adequately addressed your comments raised in a previous round of review and you feel that this manuscript is now acceptable for publication, you may indicate that here to bypass the “Comments to the Author” section, enter your conflict of interest statement in the “Confidential to Editor” section, and submit your "Accept" recommendation.

Reviewer #2: All comments have been addressed

2. Is the manuscript technically sound, and do the data support the conclusions?

Reviewer #2: Yes

3. Has the statistical analysis been performed appropriately and rigorously? 

Reviewer #2: Yes

4. Have the authors made all data underlying the findings in their manuscript fully available?

Reviewer #2: Yes

5. Is the manuscript presented in an intelligible fashion and written in standard English?

Reviewer #2: Yes

6. Review Comments to the Author

Reviewer #2: I would like to thank the authors for their revision. I can accept the selection of the examined age range (10-13 years) only from its practical aspect given the grouping method usually applied in sport clubs and academies. However, it is still not quite convincing from a research design aspect. Further, and as mentioned in my previous review, I believe biobanding should be examined in groups of similar chronological, but of different biological age (BA). As applied in this research, the differences in maturational status may be attributed to differences in chronological age (CA) (which seems reasonable). However, we still cannot clearly relate the results to either the CA or to BA. Since the aim was to examine the effects of biobanding on training load before APHV, a narrower age range could be more suitable.

7. PLOS authors have the option to publish the peer review history of their article (what does this mean?). If published, this will include your full peer review and any attached files.

Reviewer #2: No

---

## [Editor Report · Acceptance letter]

11 Jan 2025

PONE-D-24-11339R2 

PLOS ONE

Dear Dr. Laerte Lopes Ribeiro, 

I'm pleased to inform you that your manuscript has been deemed suitable for publication in PLOS ONE. Congratulations! Your manuscript is now being handed over to our production team.

Kind regards, 

on behalf of

Dr. Julio Alejandro Henriques Castro da Costa 

Academic Editor

PLOS ONE